# Effects of Human Activity on Markers of Oxidative Stress in the Intestine of *Holothuria tubulosa*, with Special Reference to the Presence of Microplastics

**DOI:** 10.3390/ijms23169018

**Published:** 2022-08-12

**Authors:** Jessica Lombardo, Antònia Solomando, Amanda Cohen-Sánchez, Samuel Pinya, Silvia Tejada, Pere Ferriol, Guillem Mateu-Vicens, Antonio Box, Caterina Faggio, Antoni Sureda

**Affiliations:** 1Department of Chemical, Biological, Pharmaceutical, and Environmental Sciences, University of Messina, 98166 Messina, Italy; 2Research Group in Community Nutrition and Oxidative Stress (NUCOX), University of Balearic Islands, 07122 Palma de Mallorca, Spain; 3Interdisciplinary Ecology Group, Department of Biology, University of the Balearic Islands, 07122 Palma de Mallorca, Spain; 4Laboratory of Neurophysiology, University of the Balearic Islands, 07122 Palma de Mallorca, Spain; 5CIBER Fisiopatología de la Obesidad y Nutrición (CIBEROBN), Instituto de Salud Carlos III (ISCIII), 28029 Madrid, Spain; 6Health Research Institute of Balearic Islands (IdISBa), 07120 Palma de Mallorca, Spain; 7Department of Agricultura, Ramaderia, Pesca, Caça i Cooperació Municipal, Consell Insular d’Eivissa, 07800 Eivissa, Spain

**Keywords:** holothurians, microplastics, biomarkers, oxidative stress, pollution, Balearic Islands

## Abstract

Pollution in the seas and oceans is a global problem, which highlights emerging pollutants and plastics, specifically microplastics (MPs), which are tiny (1 μm to 5 mm) ubiquitous plastic particles present in marine environments that can be ingested by a wide range of organisms. Holothurians are benthic organisms that feed on sediment; therefore, they can be exposed to contaminants present in the particles they ingest. The objective was to evaluate the effects of human activity on *Holothuria tubulosa* through the study of biomarkers. Specimens were collected in three different areas throughout the island of Eivissa, Spain: (1) a highly urbanized area, with tourist uses and a marina; (2) an urbanized area close to the mouth of a torrent; (3) an area devoid of human activity and considered clean. The results showed a higher presence of microplastics (MPs) in the sediments from the highly urbanized area in relation to the other two areas studied. Similarly, a higher number of MPs were observed in the digestive tract of *H. tubulosa* from the most affected area, decreasing with the degree of anthropic influence. Both in the sediment and in the holothurians, fibers predominated with more than 75% of the items. In the three areas, mesoplastics were analyzed by means of FTIR, showing that the main polymer was polypropylene (27%) followed by low-density polyethylene (17%) and polystyrene (16%). Regarding the biomarkers of oxidative stress, the intestine of *H. tubulosa* from the most impacted areas showed higher catalase, superoxide dismutase (SOD), glutathione reductase (GRd), and glutathione *S*-transferase (GST) activities and reduced glutathione (GSH) levels compared to the control area. The intermediate area only presented significant differences in GRd and GST with respect to the clean area. The activities of acetylcholinesterase and the levels and malondialdehyde presented similar values in all areas. In conclusion, human activity evaluated with the presence of MPs induced an antioxidant response in *H. tubulosa*, although without evidence of oxidative damage or neurotoxicity. *H. tubulosa*, due to its benthic animal characteristics and easy handling, can be a useful species for monitoring purposes.

## 1. Introduction

Environmental pollution causes serious imbalances in the ecosystem, with potential negative consequences for the health of living organisms. Among the pollutants, plastics have acquired great relevance due to the exponential increase in their production to meet the demands of an ever-growing population [1]. World production of plastics is about 300 million tons per year, of which at least 8 million tons accumulate in the marine ecosystem, forming the marine litter [2,3]. Plastic waste is particularly problematic as a pollutant, due to its high resistance to decomposition, persisting in the environment for a long time [4,5]. Thus, due to improper disposal and management of plastics, they accumulate in both land and water, drastically polluting the environment [6,7,8,9]. The Balearic coast is one of the main areas of accumulation of plastics in the Mediterranean Sea [10]. Diverse studies on plastics showed that 88% of the sampled seabed areas have plastic residues, and that 45% of the 40 different species sampled (including fish, mollusks, and crustaceans) have ingested microplastics [11].

Plastic materials are polymers formed following the union of simple molecules, consisting mainly of carbon molecules that are repeated to form an extremely long molecule [12]. They are not an individual entity, but constitute a cocktail of polymers and additives that can absorb and adsorb substances from the surrounding environment, including living substances, nutrients, and marine pollutants [13]. Plastic polymers, such as polyethylene (PE), polypropylene (PP), HDPE (high-density polyethylene), and LDPE (low-density polyethylene), are some of the most produced and least recycled worldwide, and they have become the most abundant in the oceans [14,15]. In this sense, PE, HDPE, and LDPE were reported to be the most prevalent plastic polymers in the Mediterranean Sea and also in the Balearic Islands [16,17,18,19,20]. Once released in the water, their environmental fate primarily depends on the polymer density, which is influenced by buoyancy, the position in the water column, and the consequent possibility to interact with biota [21]. Polymers with a density lower than that of seawater (e.g., PE and PP) will tend to float in the water column, while fragments with higher density will sink and will accumulate on the seabed [22]. Whilst entanglement and ingestion of macro debris are generally concerns for large vertebrates, MPs (less than 5 mm) can be ingested by a wide range of marine organisms, including planktonic and invertebrate organisms. Plastic fragments may accumulate biofilms, sink, and become mixed with sediment. In fact, the presence of MPs has been documented in sediments worldwide, with abundances increasing in densely populated areas [7,23,24,25,26]. Once ingested, MPs may pass through the gut or may be retained in the digestive tract where they can accumulate and clog the digestive systems, causing a false sense of satiation, but they can also cause physical tissue damage and inflammation [27,28]. In addition, during residence in the digestive tract, chemicals added during the manufacturing process or incorporated from the environment may be absorbed and cause toxic effects [18,29]. The uptake of MPs associated with environmental pollutants could act as an additional route of exposure for marine organisms to a wide array of harmful chemicals such as metals, bisphenol A, phthalates, or polycyclic aromatic hydrocarbons (PAHs) [30]. In this sense, several experimental studies have shown that the ingestion and accumulation of MPs induce the antioxidant defensive systems through the generation of reactive oxygen species (ROS) [31,32,33,34]. If ROS are not removed effectively, they can react with cellular components and cause oxidative damage to lipids, proteins, and DNA [35]. Antioxidant and detoxification enzymes play a crucial role in maintaining cell homeostasis and are extensively used as biochemical biomarkers of contaminant-induced oxidative stress in aquatic organisms [36,37]. Benthic holothurians are an adequate model for studying plastic ingestion in deposit-feeding organisms. Holothurians, as nonselective feeders, ingest large quantities of sediment into their mouths to extract nutrients from biofilms, organic debris, and microorganisms [38]. The transit through the digestive tract could allow the absorption of plastic leachates and adhered contaminants, as well as reduce the size of plastic particles. In holothurians, plastic items ingested with sand grains could be ground during their passage through the guts [39]. The presence of microplastics has been documented in many holothurians [39,40]. In this sense, Tejedor-Junco et al. (2021) confirmed the presence of microplastics in the guts of holothurians, in environments under different anthropogenic influences, with a significantly higher number in the holothurians of the polluted site, relative to the other ones [41]. *Holothuria tubulosa* (Gmelin, 1788) is a species of sea cucumber in the family Holothuriidae, widely present in coastal waters of the Balearic Islands. It can reach 30 cm in length, and it is roughly cylindrical with a flattened base on which there are three longitudinal rows of tube feet [42,43]. Like all holothurians, this species feeds by ingesting the sandy substrate; hence, it is exposed to the possible presence of plastics or contaminants in the sediment. Although the behavior of holothurians is still poorly studied, the daily distances traveled is small, which suggests that they may be suitable species as bioindicators of human activity [44].

Considering all this information, the aim of the study was to evaluate the presence and characteristics of MPs in sediments and in *H. tubulosa* sampled in three areas of the Island of Eivissa (Balearic Islands, Spain), characterized by a different anthropic influence. The antioxidant and detoxification mechanisms and cellular oxidative damage were also investigated in this species associated with the ingestion of MPs. In addition, this study provides information on the response to stress at the molecular and biochemical levels associated with contamination by MPs and determine the usefulness of *H. tubulosa* as a potential indicator organism.

## 2. Results

### 2.1. Biometric Measurements

*H. tubulosa* TL ranged from 125 to 293 mm, from 125 to 170 mm, and from 60 to 182 mm in samples collected from Sant Antoni de Portmany, Santa Eulària des Riu, and Pou des Lleó, respectively. Of the 30 specimens that were collected, mean body weight was 175.2 ± 13.4 g, ranging from 96 to 400 g. Mean intestinal weight was 65.4 g ± 12.5, ranging from 23 to 319 g.

### 2.2. Microplastic Abundance in H. tubulosa

Of the total of 30 digestive tracts that were examined, 83.3% (25 specimens) had MPs particles in their digestive tracts. A total of 104 particles (Table 1) were extracted from the digestive tracts of the 30 individuals, yielding an average of 3.5 ± 0.7 (range: 0–14) MPs per individual. Figure 1 shows representative images of the MPs observed in the gastrointestinal tracts of the animals. It is important to highlight that, in five specimens, no MP particles were detected in the digestive tract. Out of the 25 animals with MPs, 10 individuals had more MP than average (33.3%), while only two individuals presented more than 10 particles (6.7%). The number of ingested plastics differed among sampling sites. A higher number of MPs were observed in the digestive tract of *H. tubulosa* from the most polluted area, Sant Antoni de Portmany, with an average of 6.5 ± 1.6 MPs per individual. Lower values were found in the digestive tract of organisms collected from Santa Eulària des Riu and Pou des Lleó, with mean values of 2.5 ± 0.9 MPs and 1.4 ± 0.5 MPs per individual, respectively. A greater quantity of MP items was found in two organisms from Sant Antoni de Portman, with 14 particles each, although it did not correspond to the largest individuals with heavier digestive tract weight. The mean number of MPs corrected per gram of intestinal tissue was 0.43 ± 0.11, 0.17 ± 0.06, and 0.1 ± 0.03 from Sant Antoni de Portmany, Santa Eulària des Riu, and Pou des Lleó, respectively.

### 2.3. Microplastic Abundance in the Sediments of the Collection Site

A total of 137 MPs were found in the sediments obtained from the three collection sites (Table 1). The total number of MPs per studied area was 61 in Sant Antoni, 44 in Santa Eulària, and 32 in Pou des Lleó. On average, Sant Antoni de Portmany sediment showed 201 items/kg sediment. A total of 179 items/kg sediment were retrieved from Santa Eulària des Riu. Lastly, Pou des Lleó showed the lowest value with 134 items/kg sediment.

### 2.4. MPs Characterization

Regarding the characteristics of the MPs found, for *H. tubulosa*, 83.7% of the particles identified were fibers, while the remainder were small fragments. For each station, the fiber/fragment ratio obtained was 57/8 for Sant Antoni de Portmany, 20/5 for Santa Eulària des Riu, and 9/5 for Pou des Lleó. For sediment, 96.4% were fibers and 3.6% were fragments (Table 1), with a relation fiber/fragment of 52/2 for Sant Antoni de Portmany, 42/2 for Santa Eulària des Riu, and 31/1 for Pou des Lleó. As for colors, the most common MP color in all samples from *H. tubulosa* was blue (43.3%), although transparent and black (both, 13.5%) pieces were also quite abundant, as shown in Table 1. However, among all plastics collected in sediment, black (39.8%) and blue (35.8%) were the main colors, followed by red (11.4%), in terms of both number and occurrence of particles.

### 2.5. Mesoplastic Characterization

Of the mesoplastic items collected in sediment, the maximum length reported in the different study sites ranged between 4 mm (plastic fragment) and 650 mm (nylon line), with an average length of 80.0 ± 40.4 mm, which fits within the mesoplastic definition. Among the 75 plastic items classified by shape, the most abundant category was fragments with a frequency of 76%, followed by sheets (15%), and finally threads (9%). In Sant Antoni de Portmany (*n* = 25) 17 fragments (68%), five sheets (20%), and three threads (12%) were identified. In Santa Eulària des Riu (*n* = 25) and in Pou des Lleó (*n* = 25) 20 fragments (80%), three sheets (12%), and two threads (8%) per station were found. The results of each station are shown in Table 2. Among all plastics collected, blue (93%) was the most abundant color, followed by green (15%) and red (9%). Transparent (7%), yellow (7%), and gray pieces (7%) were also quite abundant, while the least abundant colors were black (5%), white (5%), orange (4%), violet (1%), and gold (1%). Table 2 also shows results of the total frequency of occurrence (FO) and total number of pieces of different colors for each station.

### 2.6. Polymer Characterization

Mesoplastic debris was characterized by ATR-FTIR spectroscopy to identify most frequent polymers. The analyses detected eight different polymers (Figure 2). Among them, the polymer with the highest frequency was polypropylene (PP; 27% of the total samples). High frequencies were also recorded of low-density polyethylene (LDPE; 17%) followed by polystyrene (PS; 16%), high-density polyethylene (HDPE; 15%), and polyvinyl chloride (PVC; 13%). Polyester was also found but in a lower proportion (PL; 8% of the total samples). Two lesser abundant polymers were also found, i.e., polyethylene terephthalate (PET; 3%) and polyamide (PA; only 1% of the total analyzed).

### 2.7. Oxidative Stress Biomarkers

Statistical analysis evidenced significant higher values in the two most impacted areas: Sant Antoni de Portmany and Santa Eulària des Riu with respect to Pou des Lleó in the gut of *H. tubulosa* of all antioxidant enzymes activities (CAT, *p* = 0.032; SOD, *p* = 0.043; GRd, *p* > 0.001, one-way ANOVA) (Figure 3), GST as a marker of the detoxification capabilities (*p* = 0.002, one-way ANOVA) (Figure 4), and total GSH (*p* = 0.018, one-way ANOVA) (Figure 5). In contrast, the activity of AChE (*p* = 0.84, one-way ANOA) (Figure 6) and the levels of MDA (*p* = 0.21, one-way ANOVA) (Figure 7), measured in the gut of *H. tubulosa*, presented similar values in all areas.

Significant differences in the activity of CAT were found between Sant Antoni de Portmany and Pou des Lleó (LSD, *p* < 0.05), with a mean value of 30.5 ± 5.4 mK/mg protein in Sant Antoni de Portmany and 18.2 ± 3 mK/mg protein in Pou des Lleó. A similar response pattern was observed in the SOD and GSH activities, evidencing significantly higher values (LSD, *p* < 0.05) in guts of animals collected from Sant Antoni de Portmany (SOD: 8.3 ± 1.5 pKat/mg; GSH: 22.1 ± 2.9 µmol/mg protein) than in the gut of animals from Pou des Lleó (SOD: 5.5 ± 0.2 pKat/mg protein; GSH: 20.1 ± 1.3 µmol/mg protein). Moreover, GST and GRd activities were significantly higher in Sant Antoni de Portmany, (GST: 23.3 ± 1.5 nKat/mg protein; GRd: 18.8 ± 4.3 nkat/mg protein) and Santa Eulària des Riu (GST: 21 ± 0.9 nKat/mg protein; GRd: 16.7 ± 3.9 nkat /mg protein) than in Pou des Lleó (GST: 16.9 ± 1.1 nKat/mg protein; GRd: 10.6 ± 2.5 nkat/mg protein) (LSD, *p* < 0.05). 

MDA levels remained similar in animals captured from the three areas, without significant differences among Sant Antoni de Portmany (36.8 ± 0.8 nM/mg protein), Santa Eulària des Riu (37.4 ± 1.3 nM/mg protein), and Pou des Lleó (35 ± 0.8 nM/mg protein) (LSD, *p* > 0.05). In the same way, no significant differences were observed between AChE activity measured in gut tissue in animals from the different stations, with a mean value of 35.5 ± 10.4 µmol/min/mg in Sant Antoni de Portmany, 35.2 ± 11.9 µmol/min/mg in Santa Eulària des Riu, and 33 ± 8.6 µmol/min/mg in Pou des Lleó.

All data obtained from enzymatic analyses and statistically significant differences are shown graphically (Figure 3, Figure 4, Figure 5, Figure 6 and Figure 7).

When analyzing the correlations between the presence of MPs in the digestive tract and the different biomarkers, direct correlations with CAT (*r* = 0.565, *p* < 0.01), GST (*r* = 0.902, *p* < 0.001), and GSH (*r* = 0.617, *p* < 0.001) were observed.

## 3. Discussion

Marine pollution is a serious problem, evident even in the most pristine and isolated environments such as Antarctica or abyssal trenches [45,46]. Among these pollutants, plastics have become a global concern due to their progressive growth and potential effects on marine ecosystems. The results obtained revealed the presence of plastic elements in the sediments of the three study areas, although their number increased with the degree of anthropic influence. In addition, these results are parallel to a greater presence of MPs in the intestinal tract of *H. tubulosa* and a greater activation of antioxidant defense mechanisms. When analyzing the presence of MPs in the sediment of the three areas, it was observed that Sant Antoni and Santa Eulàlia, areas under anthropic pressure, have higher levels than in Pou des Lleó, an area without relevant human activity. However, the presence of MPs in the area considered clean highlights the ubiquity of plastics and their high dispersion capacity. The coastal sediments of the Mediterranean Sea are among the most important sinks for the deposition of microplastics and have an immense potential for their accumulation, due to the characteristics of the semi-enclosed sea and the high population density in coastal areas [47]. The amount of MPs in the sediments was within the range of those observed in other areas of the Spanish Mediterranean with values between 45 and 280 items/kg sediment, with an average value of 113.2 ± 88.9 MPs/kg [48]. Similar values have also been found in the sediments of Andratx (Mallorca) with a number of MPs between 120 and 160 items/kg [23]. Regarding the dominant microplastic form in the sediments, our results are in agreement with the study of Filgueiras et al. (2019), where fibers predominated over fragments [48]. A predominance of fibers was also found as the most common type of MPs in surficial sediments in different areas of the Mediterranean and elsewhere [49,50,51,52]. MPs in the form of fibers mainly originate from sewage and synthetic garments from the textile industry [53]. In this sense, this type of contamination characteristic of populated areas could be attributed to sewage input rather than fragmentation of large plastic particles. When analyzing the composition of mesoplastics, the main polymers found were polypropylene, polyethylene, and polystyrene, which derive from the most demanded thermoplastics for packaging, construction materials, and electrical apparatus [54,55].

The abundance of microplastics inside some holothurians (i.e., *A. japonicus* and *H. tubulosa*) is generally lower than in sediments [50,56], suggesting that most of the ingested items are incorporated into fecal pellets and expelled. However, Graham and Thompson (2009) [38] observed that four species of deposit-feeding and suspension-feeding sea cucumbers ingested significantly more plastic fragments than predicted given the ratio of plastic to sand grains in the sediment. In the present study, the presence of MPs in the digestive tract was quite similar to that observed in the sediment, although, in the case of the area with more MPs in the sediment, the values in sea cucumbers were somewhat higher. Even so, the amount of MPs in the gastrointestinal tract varied in a similar way to those found in the sediments, suggesting that this species can be a good indicator of contamination by plastics. Moreover, although sea cucumbers feed unselectively by ingesting sandy substrate, there is some evidence to suggest that they may actively take some MPs types over others with a slight preference for plastic fragments over other plastic shapes [38,56]. This characteristic could explain the higher percentage of fibers in the digestive tract of *H. tubulosa* than in the sediments. In the present study, the main colors in the MPs analyzed in the sediment and *H. tubulosa* were blue and black. The colors of the MPs differ between different studies, although there is a predominance of the blue color, due to its high use in packages, bottles, and caps [57,58,59]. The color distribution is probably due to the different origin of the plastic materials in the area studied or to the degradation processes to which they are subjected in the marine environment [58]. 

The effects of ingestion of MPs in marine organisms have been extensively studied in a wide variety of marine organisms from plankton [60], to bivalves [61], crustaceans [62], fish [63,64], marine turtles [14], and cetaceans [65]. Previous studies have found MPs in the tissues of different holothurians species, suggesting a potential transfer to upper trophic levels [38,50,51]. However, studies linking the presence of MPs in holothurian species with their potential physiological or biochemical effects are still scarce. The present results evidenced that the antioxidant enzyme activities—CAT, SOD, and GRd—and the levels of GSH and GST activity progressively increased in accordance with the degree of human impact. In this sense, oxidative status biomarkers are deemed to be useful to identify the effects of environmental stressors effects in marine organisms [66,67]. To cope with the toxic effects of MPs and potentially absorbed contaminants or other toxic compounds, organisms dispose an elaborate ROS elimination system. Previous studies have reported the potential of MPs to induce the production of ROS followed by a disruption in the antioxidant defense systems in diverse organisms, mostly in fish and bivalves [68,69,70]. In relation to studies carried out in holothurians, these are scarce and were developed mainly in controlled laboratory conditions that generally do not reflect environmental settings and expose the organisms to concentrations of pollutants higher than the real ones [71,72]. However, the data obtained in one of these studies evidenced that the exposure of *Apostichopus japonicus* (Selenka) to dietary polystyrene nanoplastics (NPs) or MPs induced an increase in ROS production and MDA content in coelomic fluid and depressed cellular and humoral immune parameters [71]. In addition, after 60 days, the activities of CAT and SOD were reduced in the NPs group but not in the MPs group, suggesting the existence of size-related toxicity differences between NPs and MPs. In another study, the short-term exposure of *A. japonicus* to polyester MPs for 72 h did not evidence significant differences in the antioxidant/prooxidant biomarkers in the coelomic fluid, while the levels of lysozyme were increased [50]. The absence of significant changes in the antioxidant response in these studies may be due to relatively short exposure times, compared to the chronic exposure that occurs under natural conditions and also because virgin plastics without other associated contaminants were used. In this sense, significant increases in antioxidant defense mechanisms and GST have been observed in the digestive tract of *Holothuria forskali* after exposure to mercury [72]. Thus, the effects observed in the present study may indicate the combined effect of different pollutants, including MPs, associated with human activity. It is interesting to note that the observed increase in GSH levels was probably due to a more oxidized redox state as evidenced by an increase in GRd activity, as well as its consumption associated with GST activity [73].

The induction of antioxidant defense mechanisms has as main objective to maintain the homeostasis of the organism and avoid the appearance of oxidative damage [34]. In the present study, changes in antioxidant enzymes in the most polluted areas were sufficient to prevent increases in MDA levels. However, previous studies have observed an increase in ROS levels and oxidative damage markers in *A. japonicus* exposed to MPs, benzo[a]pyrene (BaP), or crude oil under controlled conditions [71,74,75]. Similarly, no significant changes in AChE activity were observed between the areas studied, indicating the absence of neurotoxicity. Exposure to pollutants can induce neurotoxicity usually evidenced by an inhibition of AChE activity, such as that observed in *H. forskali* exposed to increasing levels of mercury [72].

The changes observed in the MP content in the intestinal tract of *H. tubulosa* and its direct relationship with the environmental presence of plastics, in addition to the response of the analyzed biomarkers, suggest that this species may be a suitable bioindicator for coastal pollution studies. There is the limitation that additional contaminants to the MPs have not been determined; thus, the observed changes cannot be attributed exclusively to the MPs. However, this is the first study to investigate the response of *H. tubulosa* to human activity and its usefulness for monitoring environmental changes.

In conclusion, the results revealed the presence of MPs in the digestive tract of *H. tubulosa* specimens, even in those captured in areas without human activity. The amount of MPs in the digestive tracts varied in a similar way to the presence of plastics in the sediment; hence, they can be indicative of the degree of anthropic impact. In addition, an increase in antioxidant defense and detoxification mechanisms was observed as human influence increased but without evidence of oxidative damage and neurotoxicity. The use of oxidative stress biomarkers is a good proxy to evaluate or monitor the effects of pollution arising from potential toxic effects of MPs in *H. tubulosa*. In this sense, *H. tubulosa* may be a suitable organism for environmental pollution monitoring studies. Plastics will continue to shape our present and our future; however, we will not be able to achieve the full potential of these materials if we do not address the global challenges linked to their negative impact when they end up in the environment.

## 4. Materials and Methods

### 4.1. Holothuria Sampling and Study Area

Coastal ecosystems are subject to human pressures such as acting as pollution sinks, especially in sparsely populated areas, whose presence represents a recognized environmental threat. In this sense, Eivissa Island is also strongly influenced by tourism, especially during summer months, when the population exceeds 152,000 inhabitants [76]. Since plastic waste has become one of the most studied indicators of anthropogenic impact, plastic pollution and its effects on *H. tubulosa* were investigated in three different areas of the island. A total of 30 *H. tubulosa* individuals were collected throughout the Island of Eivissa (Spain), located in the Western Mediterranean (Figure 8), during April 2022. Samplings were conducted at three different stations with different levels of anthropogenic influences: (1) a highly urbanized area, with tourist uses and a marina, Sant Antoni de Portmany; (2) an urbanized area close to the mouth of a torrent, Santa Eulària des Riu; (3) an area devoid of human activity and considered clean, Pou des Lleó. 

Specimens were collected by handpicking using a mask and snorkel at depths of 1–2 m. Immediately after the catch, the specimens were carefully cleaned of attached marine invertebrates, seaweeds, and sand in the field [77]. Thereafter, the holothurians were anaesthetized with clove powder to minimize stress and frozen at −20 °C. In the laboratory, individuals were measured obtaining the total length (TL; ±0.1 cm) and weight (wet weight; ±1 g). Then, the sea cucumbers from each site were eviscerated, and the whole digestive tract (from top of the esophagus through the end of the gastrointestinal tract) and gut samples were removed and frozen at −80 °C, until their analyses. 

Sediment samples were randomly obtained from each study area, at the same depth as *H. tubulosa* specimens (1–2 m), with three replicates from each site. An auger was employed to collect the sediment, inserting it into the sediment surface at a 0°–45° angle from vertical to minimize the loss of surface sediment and water entering. Because MPs were being sampled, there was no concern in mixing the sediment, which was then stored in clean glass and prelabeled containers [78].

### 4.2. Mesoplastic Characterization

During the transects carried out to collect the holothurians, mesoplastics present in the sandy bottom were randomly collected to proceed to their characterization and composition analysis by FTIR. A total of 75 plastic items found (25 for each station) were counted, measured (maximum length), and categorized according to shape, color and type of polymer. Plastic items were classified by shape as fragments, sheet-like, and thread-like; they were also classified into the 11 observed color categories (blue, green, red, transparent, yellow, grey, black, white, orange, violet, and gold). Mesoplastics sampled were characterized by FTIR (Bruker OPTICS, Ettlingen, Germany) to identify common polymers. Before FTIR analysis, plastic items were carefully rinsed with deionized water and dried, with the purpose of achieving precise spectra [79]. FTIR measurements were conducted in attenuated total reflectance (ATR) mode using a wave number range between 400 and 4000 cm^−1^, 16 coadded scans, and a spectral resolution of 4 cm^−1^. The spectra generated were subjected to baseline correction in order to lessen noise and improve the spectrum quality. All spectra were then compared with commercial and custom-made spectral databases. Similarities greater than a 70% hit quality index (HQI) were considered acceptable [80]. The analysis was carried out with the support of the Scientific/Technical Services at the University of the Balearic Islands.

### 4.3. Microplastic Ingestion

The MP extraction protocol from *H. tubulosa* was performed as previously described with some modifications [81,82]. Briefly, prior to chemical digestion, all digestive tracts were left at room temperature to defrost. Then, 15 g of soft tissue in order to homogenize size differences was introduced into a properly labeled glass Erlenmeyer flask and incubated with 10% potassium hydroxide (KOH) (20 mL of KOH/g tissue). The chemical digestion took place for 48–72 h at 60 °C, and all flasks were covered with aluminum foil to avoid air contamination. The digested solution was filtered through a vacuum filter using polycarbonate filters (FILTER-LAB polycarbonate membrane filters, pore size 1.2 μm, diameter 47 mm, Prat Dumas, France) inside a fume hood to avoid airborne contamination. The filter discs were removed and individually placed inside glass Petri dishes and allowed to dry at room temperature for 24 h. Finally, filters were observed under a stereomicroscope (Leica EZ4) for MP visual identification. MP items were counted per specimens, and color and shape were described. The MP shape was categorized into two groups: “fibers” (items slender or elongated in shape) and “fragments” (items angular, flat in shape) [83]. Images of MP particles identified were taken using a Leica DFC295 digital camera (optical enhancement up to 11.5×) and Leica application suite software.

For the sediment analysis in the laboratory, 250 g of sediment from the collection site was mixed with a previously filtered high-density saline solution (1 L H_2_O + 120 g NaCl, 1.2 g·cm^−3^ NaCl) to generate a solution that allowed the MPs to float [84]. The mix was stirred for 2 min, and the sand particles were allowed to settle for 1 h. Then, the saline solution with floating MPs was filtered and dried, and MPs were visually identified using the same methodology as for the digestive tracts. The total number of MPs obtained in each studied area was also corrected for kg of sediment.

To prevent contamination during field and laboratory analyses, all instruments were washed thrice with pure water (Milli-Q^®^ A10 Direct Water Purification System, Merck KGaA, Darmstadt, Germany). In addition, cotton lab coats and nitrile gloves were worn throughout experimental procedures, and blank controls using distilled water were also visually inspected under the stereomicroscope to detect any possible airborne plastic contamination. Moreover, all saline solutions used were previously filtered to remove possible microparticles.

### 4.4. Biomarkers

To determine the possible effects of the MP ingestion on the instauration of oxidative stress, several biomarkers were measured in the gut. SOD, CAT, and GRd activities and GSH levels, as antioxidant defenses and, GST activity, as an enzyme implicated in detoxification processes were determined. MDA levels as marker of oxidative damage to lipids and AChE as an indicator of potential neurotoxicity were also evaluated. Gut samples were homogenized under ice-cold conditions in 10 volumes (*w*/*v*) of 100 mM Tris–HCl buffer pH 7.5 using a small sample dispersing system (Ultra-Turrax^®^ T10 Disperser, IKA, Staufen, Germany) and centrifuged (9000× *g*, for 10 min, 4 °C; Sigma 3K30) [35]. After centrifugation, supernatants were collected and used for all the biochemical analyses.

SOD activity (pKat/mg protein) was determined according to Flohé & Ötting (1984), at 550 nm [85]. CAT activity (mK/mg protein) was measured as previously described by Aebi (1984) at 240 nm [86]. GRd activity (nmol/min/mg protein) was measured by a modification of the Goldberg and Spooner method at 339 nm [87]. GST activity (nKat/mg protein) was determined at 340 nm using the technique of Habig et al. (1974) [88]. The activity of AChE (µmol/min/mg) was determined at 412 nm, using the method of Ellman et al. (1961) with slight modifications [89]. SOD, CAT, GRd, GST, and AChE activities were determined using a Shimadzu UV-2100 spectrophotometer at 25 °C. GSH levels (nmol/mg prot) were measured at 415 nm in a microplate reader (Bio-Tek^®^ PowerWaveXS, Agilent Technologies, Madrid, Spain) following an adaptation of the method described by Pinya et al. (2016) [90]. MDA levels (nM/mg protein) were quantified by a colorimetric assay kit (Merck, Madrid, Spain), following the manufacturer’s instructions.

All results were referred to the total protein content of the samples determined with the colorimetric Bradford method (Biorad^®^ Protein Assay, Alcobendas, Spain) using bovine serum albumin as a standard.

### 4.5. Statistical Analysis

Data obtained from the experiment were statistically analyzed using a statistical package (SPSS 25.0 for Windows^®^, IBM^®^ SPSS Inc., Chicago, IL, USA). The normality of distribution and equality of variance of the data were evaluated using the Kolmogórov–Smirnov test and Levene’s test, respectively. Then, the statistical significance of the differences in all the biomarkers was determined through one-way analysis of variance (ANOVA). In addition, a least significant difference *t*-test (LSD) post hoc analysis was conducted to determine the statistical differences in biological parameters between studied areas. Results are expressed as the mean ± standard error of the mean (SEM), and a *p*-value < 0.05 was considered statistically significant. Bivariate correlations between the MP levels and the different biomarkers were also analyzed through Pearson correlation.

## Figures and Tables

**Figure 1 ijms-23-09018-f001:**
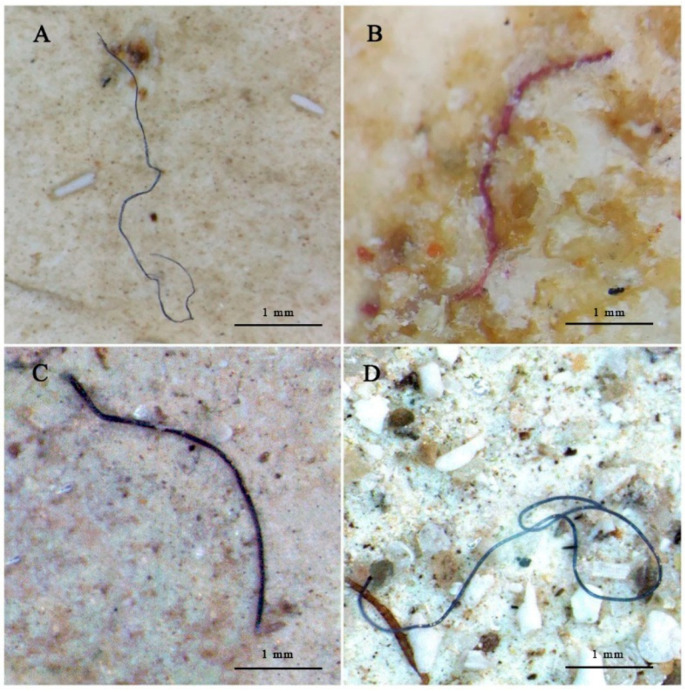
Representative MP particles found in the gastrointestinal tract of *H. tubulosa* (**A**,**B**) and representative MP particles found in the sediments (**C**,**D**): (**A**) blue fiber particle; (**B**) pink fiber particle; (**C**) black fiber particle; (**D**) black fiber particle. Scale bar represents 1 mm.

**Figure 2 ijms-23-09018-f002:**
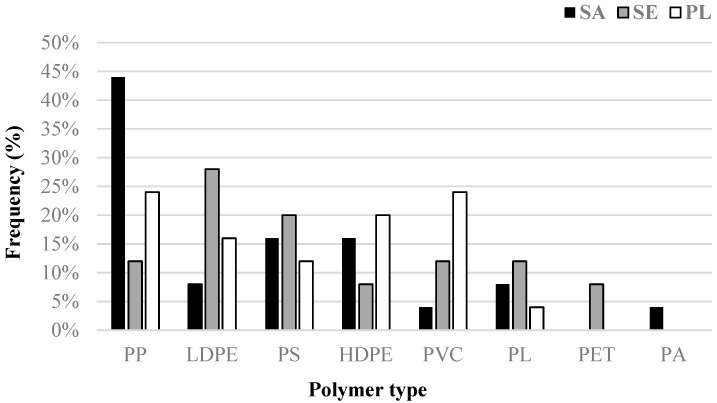
Frequency (%) of plastic polymers found by FTIR analysis (*n* = 75) of plastic items collected in the three stations (PP: polypropylene, LDPE: low-density polyethylene, PS: polystyrene, HDPE: high-density polyethylene, PVC: polyvinyl chloride, PL: polyester, PET: polyethylene terephthalate, PA: polyamide). Stations names are abbreviated as follows: SA (Sant Antoni de Portmany), SE (Santa Eulària), PL (Pou des Lleó).

**Figure 3 ijms-23-09018-f003:**
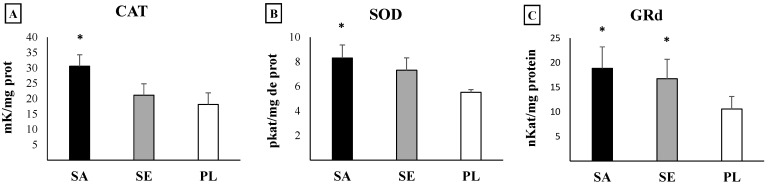
Activities of antioxidant enzymes in the digestive tract of *H. tubulosa*: (**A**) catalase (CAT); (**B**) superoxide dismutase (SOD); (**C**) glutathione reductase (GRd). Significant differences (*p*-value < 0.05) are shown: * indicates differences respect to PL. SA, Sant Antoni de Portmany; SE, Santa Eulària des Riu; PL, Pou des Lleó.

**Figure 4 ijms-23-09018-f004:**
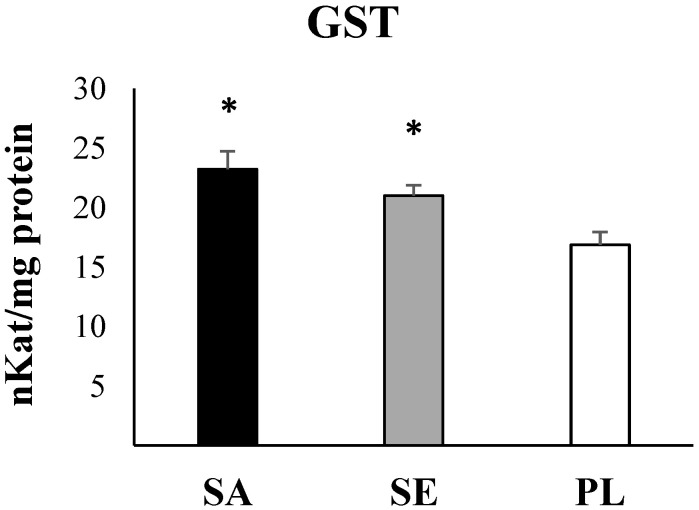
Activity of glutathione *S*-transferase (GST) in the digestive tract of *H. tubulosa*. Significant differences (*p*-value < 0.05) are shown: * indicates differences respect to PL. SA, Sant Antoni de Portmany; SE, Santa Eulària des Riu; PL, Pou des Lleó.

**Figure 5 ijms-23-09018-f005:**
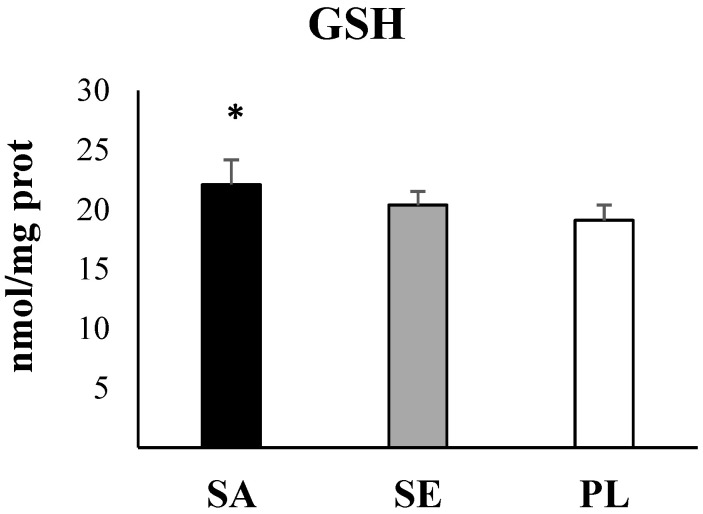
Levels of reduced glutathione (GSH) in the digestive tract of *H. tubulosa*. Significant differences (*p*-value < 0.05) are shown: * indicates differences respect to PL. SA, Sant Antoni de Portmany; SE, Santa Eulària des Riu; PL, Pou des Lleó.

**Figure 6 ijms-23-09018-f006:**
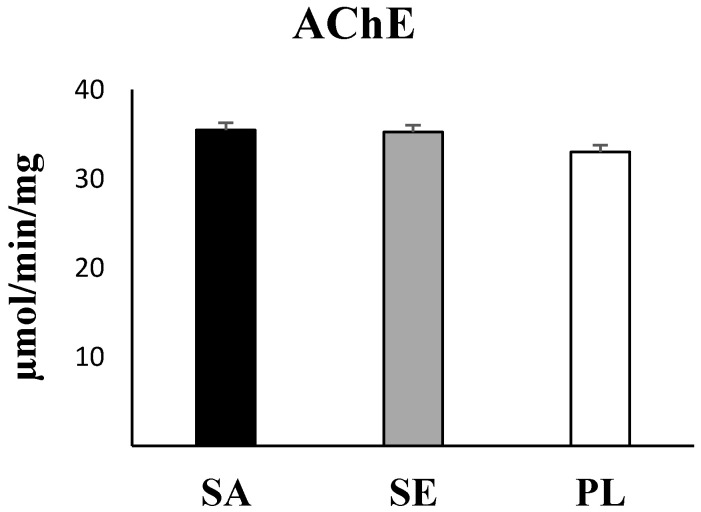
Acetylcholinesterase (AChE) activity in the digestive tract of *H. tubulosa*. No significant differences were observed. SA, Sant Antoni de Portmany; SE, Santa Eulària des Riu; PL, Pou des Lleó.

**Figure 7 ijms-23-09018-f007:**
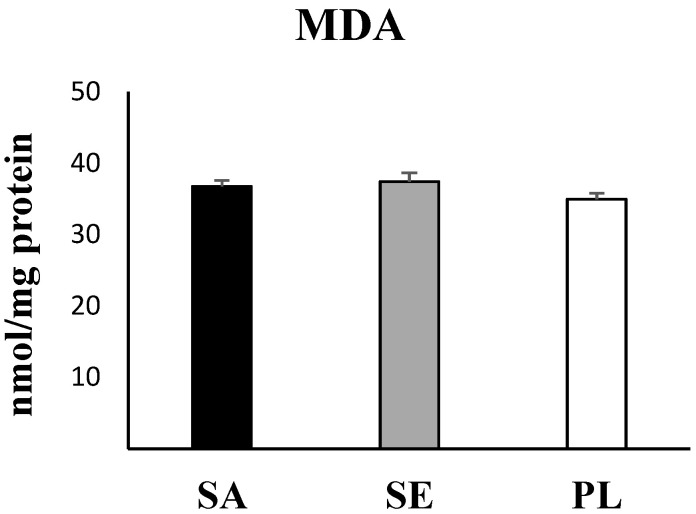
Malondialdehyde (MDA) levels in the digestive tract of *H. tubulosa*. No significant differences were observed. SA, Sant Antoni de Portmany; SE, Santa Eulària des Riu; PL, Pou des Lleó.

**Figure 8 ijms-23-09018-f008:**
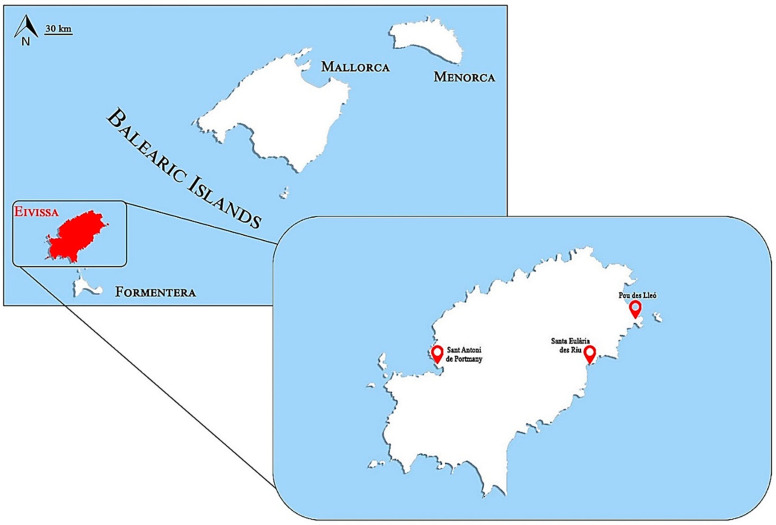
Map of the geographic localization of Balearic Islands with the names of the four principal islands. Eivissa Island is indicated in red and, in the inset map, the areas of the three sampling stations (Sant Antoni de Portmany, Santa Eulària des Riu, and Pou des Lleó) are indicated.

**Table 1 ijms-23-09018-t001:** Total number of MPs in the studied areas in *H. tubulosa* and sediments and frequency of occurrence (FO) of different shape and color.

*H. tubulosa* (*N* = 104)		*Sediments* (*N* = 137)	
Shape	FO (%)	Shape	FO (%)
Fibers	83.7%	Fibers	96.4%
Fragments	16.30%	Fragments	3.6%
Color	FO (%)	Color	FO (%)
Blue	43.3%	Black	39.8%
Transparent	13.5%	Blue	35.8%
Black	13.5%	Red	11.4%

**Table 2 ijms-23-09018-t002:** Total number (*n* = 75) and frequency of occurrence (FO) of differently shaped and colored mesoplastic items recovered of the three stations: Sant Antoni de Portmany, Santa Eulària, and Pou des Lleó.

	Sant Antoni de Portmany (*n* = 25)	Santa Eulària des Riu (*n* = 25)	Pou des Lleó (*n* = 25)	
Shape				FO (%)
Fragments	17	20	20	76%
Sheet	5	3	3	15%
Thread	3	2	2	9%
Color				FO (%)
Blue	9	8	12	39%
Green	4	3	4	15%
Red	2	2	3	9%
Transparent	2	2	1	7%
Yellow	1	1	3	7%
Grey	1	2	2	7%
Black	2	2	0	5%
White	1	3	0	5%
Orange	1	2	0	4%
Violet	1	0	0	1%
Gold	1	0	0	1%

## Data Availability

Requestors wishing to access the trial data used in this study can make a request to the corresponding author.

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
