# Peer review of "Effects of Human Activity on Markers of Oxidative Stress in the Intestine of *Holothuria tubulosa*, with Special Reference to the Presence of Microplastics"

_ijms, 2022, doi:10.3390/ijms23169018_

Round 1

Reviewer 1 Report

Review on Manuscript Number: ijms-1853939-peer-review-v1

Title: “Effects of human activity on markers of oxidative stress in the intestine of Holothuria tubulosa, with special reference to the presence of microplastics”

Comments for the manuscript

1. Summary

This manuscript presents a study on the evaluation of the occurrence and characteristics of microplastics in sediments and in Holothuria tubulosa. The study area was the Island of Eivissa, in the Balearic Islands, Spain, where different anthropic impacts were identified. The antioxidant and detoxification mechanisms and cellular oxidative damage associated to the ingestion of MPs were studied. The response to stress at the molecular and biochemical levels, associated with contamination by MPs, was evaluated to determine the usefulness of H. tubulosa as a potential indicator organism of anthropic pollution.

2. Overall opinion

The subject addressed in the manuscript is of interest to researchers and decision makers. Microplastics are a well known major current environmental problem, in particular in the marine aquatic media, with indirect impact in human health. The authors evaluated the potentiality of Holothuria tubulosa as an indicator organism for environmental pollution monitoring studies, in particular for monitor the potential toxic effects of MPs.

The manuscript is well structured and organized, and there is an easy flow between the results, discussion and conclusion. In the final part, the methodology is described in a clearly mode. Data was subject to statistical analysis. The presentation of data, and discussion are well founded and illustrated with appropriate tables and figures. The major findings are presented in an objective mode.

So, in this context, I consider that the research study presented in the manuscript meet the publishing objectives of IJMS and is adequate to be published in this scientific journal.

3. Major comments

It would be interesting, and will improve the manuscript, to include the results on the ANOVA and bivariate correlation analysis, mentioned in section 4.5.

Line 386, At which depth of water the sediment samples were collected?

The text needs editing mainly to correct typos in the text.

4. Minor comments

Lines 114-115, “The antioxidant and detoxification mechanisms and cellular oxidative damage were also in this species associated to the ingestion of MPs.” This sentence needs to be clarified. Does the authors mean that “antioxidant and detoxification mechanisms and cellular oxidative damage” were investigated?

Line 153, “Scale bar represents 1 mm.” This information should be included in the images.

Line 254, [45,46]

Line 284, Graham & Thompson (2009) – citation format

Line 315 to 318, the mentioned studies has to be cited

Author Response

  1. Major comments

It would be interesting, and will improve the manuscript, to include the results on the ANOVA and bivariate correlation analysis, mentioned in section 4.5.

The information about the results of the ANOVA and bivariate correlations have been incorporated in the revised version of the manuscript.

Line 386, At which depth of water the sediment samples were collected?

Sediment samples were collected at the same depth as H. tubulosa specimens (1-2 meters). This information has been added to the text.

The text needs editing mainly to correct typos in the text.

The text has been carefully revised in order to correct typos.

  1. Minor comments

Lines 114-115, “The antioxidant and detoxification mechanisms and cellular oxidative damage were also in this species associated to the ingestion of MPs.” This sentence needs to be clarified. Does the authors mean that “antioxidant and detoxification mechanisms and cellular oxidative damage” were investigated?

The reviewer is correct in the comment as the sentence is incomplete. In the new version "we were investigated" has been added.

Line 153, “Scale bar represents 1 mm.” This information should be included in the images.

The images were modified in order to incorporate the required information about the scale bar.

Line 254, [45,46]

It has been corrected.

Line 284, Graham & Thompson (2009) – citation format

The reference number has been added to the text “[38]”.

Line 315 to 318, the mentioned studies has to be cited.

The references have been mentioned in the revised version.

Reviewer 2 Report

General Comments

The authors present a study evaluating microplastic contamination in marine environments in relation to accumulation within the gut of benthic organisms. The results of the study suggest that microplastic accumulation both in benthic organisms and sediment increases with increased anthropogenic activity. In addition, the authors evaluated the effects of microplastic accumulation on enzyme activity within Holothuria tubulosa with respect to reactive oxygen species and neurotoxicity. While enzyme activity did increase for some of the enzymes evaluated, it is not clear if this is due to chronic MP exposure or combined MP exposure and the presence of other pollutants within the habitats under study. The authors also did not observe any increase in acetylcholinesterase activity, suggesting little neurotoxicity associated with the study locations.

Overall, the manuscript is well-written and the data collected support the conclusions drawn by the authors. While I do not completely agree that the results suggest H. tubulosa and enzyme activity may be useful biomarkers for MP pollution, it is clear that more research of this study system is warranted. Therefore, it is recommended that the manuscript be accepted for publication after minor revision. Specific comments are below.

Specific Comments

Line 27: “Holothuria tubulosa” should be italicized

Line 33: “H. tubulosa” should be italicized

Line 37: “H. tubulosa” should be italicized

Line 43: “H. tubulosa” should be italicized

Line 44: “H. tubulosa” should be italicized

Line 63: “…form extremely long molecule” should be “form an extremely long molecule.”

Line 110: “…still poor studied,…” should be “…still poorly studied,…”

Line 144: Are there error estimates that are associated with the mean number of MPs corrected per gram of intestinal tissue?

Lines 147-148: The table header should be revised. It currently reads as if the data presented will be separated by location. Instead, the data are totaled across all sites. Is there a reason for not separating the data by location? If one of the objectives is to look at the influence of anthropic influence on microplastic pollution, it would seem more relevant to separate these data by location.

Lines 159-161: It is not clear how the authors arrived at these values. If there were only 137 total MPs found across all sites, how is it possible to find 179 items/kg of sediment? It is also unclear from the methods what mass of sediment was searched for MPs. If these values were obtained by taking the average number of MPs per mass of sediment and correcting it to kg of sediment, this should be more clearly stated in the methods.

Line 163: “H. tubulosa” should be italicized

Line 205: “…respect to…” should be “…with respect to…”

Line 215: Only one value is reported for the GST and GRd activities of Sant Antoni de Portmany

Line 228-230: Is it possible to include these correlations as figures in the manuscript? It would be relevant for the reader to see these correlations.

Lines 270-271: The source of this information should be cited.

Line 325: “…did no evidence…” should be “…did not evidence…”

Line 333: “PMs” should be “MPs”

Line 361: “H. tubulosa” should be italicized

Line 372: “H. tubulosa” should be italicized

Line 373: “Holothuria tubulosa” should be italicized

Author Response

Specific Comments

Line 27: “Holothuria tubulosa” should be italicized

The scientific name has been italicized.

Line 33: “H. tubulosa” should be italicized

Corrected.

Line 37: “H. tubulosa” should be italicized

Corrected.

Line 43: “H. tubulosa” should be italicized

Corrected.

Line 44: “H. tubulosa” should be italicized

Corrected.

Line 63: “…form extremely long molecule” should be “form an extremely long molecule.”

The sentence has been corrected.

Line 110: “…still poor studied,…” should be “…still poorly studied,…”

The sentence has been corrected.

Line 144: Are there error estimates that are associated with the mean number of MPs corrected per gram of intestinal tissue?

The error of the MPs corrected per gram of intestinal tissue has been added to the revised version of the manuscript.

Lines 147-148: The table header should be revised. It currently reads as if the data presented will be separated by location. Instead, the data are totaled across all sites. Is there a reason for not separating the data by location? If one of the objectives is to look at the influence of anthropic influence on microplastic pollution, it would seem more relevant to separate these data by location. At first we had not separated by stations because in the case of the Pou des Lleó the number of microplastics was very low and could give distorted values that did not fully reflect reality. However, considering the reviewer's comment we have added the required information in the text.

Lines 159-161: It is not clear how the authors arrived at these values. If there were only 137 total MPs found across all sites, how is it possible to find 179 items/kg of sediment? It is also unclear from the methods what mass of sediment was searched for MPs. If these values were obtained by taking the average number of MPs per mass of sediment and correcting it to kg of sediment, this should be more clearly stated in the methods.

Indeed, as the reviewer comments, the values were obtained by taking the average number of MPs per mass of sediment and correcting it to kg of sediment. For a better understanding, this fact has been clarified in the methodology section.

Line 163: “H. tubulosa” should be italicized.

Corrected.

Line 205: “…respect to…” should be “…with respect to…”

The sentence has been revised and corrected.

Line 215: Only one value is reported for the GST and GRd activities of Sant Antoni de Portmany.

The reviewer is right in the appreciation; in the revised version of the manuscript the missing information has been added to the text.

Line 228-230: Is it possible to include these correlations as figures in the manuscript? It would be relevant for the reader to see these correlations.

We appreciate the comment of the reviewers, however the authors think that adding new figures would add complexity to the manuscript and would not provide additional information. However, if in a second round, the reviewer considers that it is necessary to improve the manuscript, we could incorporate the figures of the correlations.

Lines 270-271: The source of this information should be cited.

The required references were cited in the text.

Line 325: “…did no evidence…” should be “…did not evidence…”

It has been corrected.

Line 333: “PMs” should be “MPs”

The typo has been corrected.

Line 361: “H. tubulosa” should be italicized

Corrected.

Line 372: “H. tubulosa” should be italicized

Corrected.

Line 373: “Holothuria tubulosa” should be italicized

Corrected.